# FACTGUARD: DETECTING UNANSWERABLE QUESTIONS IN LONG-CONTEXT TEXTS FOR RELIABLE LLM RESPONSES

## ABSTRACT

Large language models (LLMs) have demonstrated significant advances in reading comprehension. However, a persistent challenge lies in ensuring these models maintain high accuracy in answering questions while reliably recognizing unanswerable queries. This issue remains critical, particularly as the length of supported contexts continues to expand. To address this challenge, we propose a collaborative multi-task workflow called FactGuard to automatically generate evidence-based question-answer pairs and systematically construct unanswerable questions. Using this methodology, we developed the FactGuard-Bench dataset, which comprises 25,220 examples of both answerable and unanswerable question scenarios, with context lengths ranging from 4K to 128K. Experimental evaluations conducted on nine popular LLMs reveal that all LLMs exhibit significant performance gap between answerable and unanswerable questions and the most advanced models achieve only 67.67% overall accuracy. After training with FactGuard-Bench, the model achieves an overall accuracy of 81.17%, along with enhanced reasoning capabilities on unanswerable questions. Our code is publicly available at https://anonymous.4open.science/r/FACTGUARD-5BBC

## 1 INTRODUCTION

Comprehending text and answering questions are foundational capabilities in the field of Natural Language Processing (NLP). Over the years, large language models (LLMs) have made substantial progress in reading comprehension, including the ability to process long-context inputs of up to 128K tokens(Yang et al., 2025; Liu et al., 2024). However, LLMs often tend to be overconfident (Slobodkin et al., 2023) and specially face an increased risk of generating hallucination or plausible content on unanswerable questions Deng et al. (2024). This will undermine confidence in LLM capabilities and diminish their overall reliability.

Extracting answers to answerable questions or providing justifications for why certain questions are unanswerable is equally essential for enhancing the practicality of LLMs. Answerable questions can be resolved using information contained in the provided context, while unanswerable questions arise when the context lacks sufficient or reliable evidence to support a definitive response. Handling unanswerable questions presents a particularly challenging scenario, as it requires LLMs to deeply comprehend the context, accurately determine that the question cannot be answered, and provide appropriate reasons to convince the user.

Recently, many advanced works have made a lot of efforts on unanswerable questions (Deng et al., 2024; Yehuda et al., 2024; Rajpurkar et al., 2018). SQuAD 2.0 (Rajpurkar et al., 2018) focuses on the reading comprehension of models in both answerable and unanswerable questions with manual annotation. Its texts are constrained by a context length of under 4K tokens and it does not include explicit refusal responses for unanswerable questions. SelfAware (Yin et al., 2023) employs a straightforward approach that prompts LLMs to detect unanswerable questions and response to them using predefined replies such as, "The answer is unknown". KUQ (Amayuelas et al., 2024) handles open-source LLMs on Known-Unknown questions in open-ended question-answering scenarios rather than questions related to reading comprehension. Self-Aligned method(Deng et al., 2024)

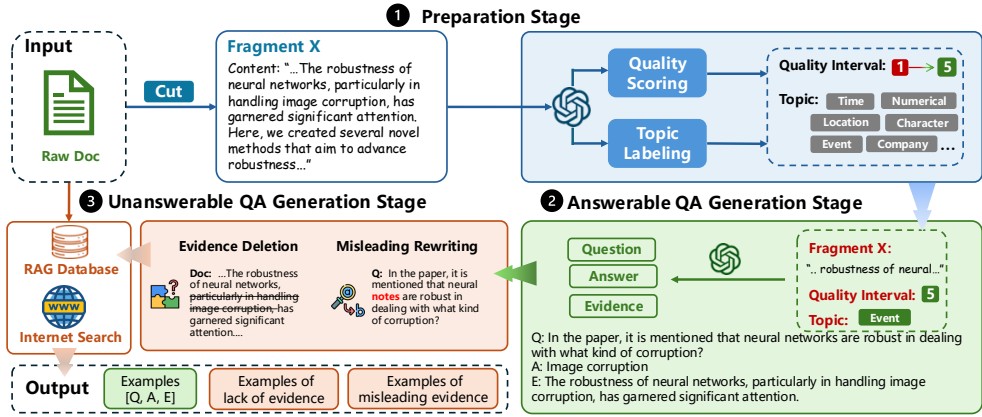

Figure 1: Illustration of FactGuard for data synthesis in a collaborative multi-task workflow framework.

mainly focuses on reasoning responses to unanswerable questions and does not pay attention to long context and it also requires manually labeled questions as seed data.

To overcome the above limitations, we propose a novel approach that employs a collaborative multi-task workflow framework to enable automated data augmentation. We introduce **FactGuard-Bench**, a reading comprehension dataset of 25,220 questions (8,829 answerable and 16,391 unanswerable) with context lengths ranging from 4K to 128K, developed through our framework. Experiments show that even the best-performing model achieves an overall accuracy of 67.67%, with significantly lower performance on unanswerable questions compared to answerable ones. Through further training, we achieved an overall accuracy of 81.17%, along with enhanced reasoning capabilities on unanswerable questions.

We highlight our contributions as follows:

1. **Innovative collaborative multi-task workflow for Data Augmentation**: We introduce **FactGuard**, a collaborative multi-task workflow framework designed to dynamically generate both answerable and unanswerable questions through a coordinated multi-step process. This approach produces contextually challenging examples that enhance the comprehension ability of LLMs.

2. **Development of Benchmark for Long-Context Evaluation**: We curate **FactGuard-Bench**, a long texts benchmark specifically tailored to assess the ability of LLMs to handle answerable and unanswerable questions.

3. **Enhancement of LLMs on unanswerable questions**: Experiments with state-of-the-art LLMs show our method enhanced the model's ability to handle unanswerable questions and generated well-reasoned answers when solving unanswerable questions.

## 2 RELATED WORK

### 2.1 MACHINE READING COMPREHENSION

Machine reading comprehension (MRC) is a hot research topic in the field of NLP, which focuses on reading documents and answering related questions (Liu et al., 2019; Baradaran et al., 2022). Over the years, machine reading comprehension has garnered significant attention from both academia and industry (Hermann et al., 2015; Liu et al., 2019). With the rapid advancements of large language models (LLMs) Zhao et al. (2023); Liu et al. (2023), retrieval-augmented generation (RAG) has emerged as a promising framework for tackling reading comprehension tasks across diverse specialized domains (Zhao et al., 2024; Lewis et al., 2020). Nevertheless, even state-of-the-art RAG frameworks are susceptible to retrieval accuracy limitations (Hu et al., 2019; Wang et al., 2025), which emphasizes the critical importance of factity Jacovi et al. (2025); Bi et al. (2025), i.e., the

ability of a model to generate factually consistent and verifiable responses in information-seeking scenarios. In this work, we emphasize scalable and robust evaluation of answerable and unanswerable questions in reading comprehension.

## 2.2 LONG CONTEXT LLMS AND BENCHMARKS

Recent studies have emphasized the importance of extending positional embeddings to improve the ability of LLMs to handle long contexts effectively (Su et al., 2024; Press et al., 2021; Chi et al., 2022). Closed-source LLMs, in particular, have emerged as leaders in long-context modeling, benefiting from progressively larger context windows. For instance, models such as GPT-4 (Achiam et al., 2023) and Gemini Pro 1.5-1000k (Team et al., 2024) are capable of processing increasingly longer documents, with context lengths ranging from 128k to 1000k tokens. Similarly, open-source LLMs, including Qwen 2.5 (Yang et al., 2024a) and DeepSeek (DeepSeek-AI, 2024), also support context lengths of at least 128k tokens. Key benchmarks for assessing long-context capabilities include NIAH (gkamradt, 2023; Yu et al., 2025), Longbench Series (Bai et al., 2024; 2025), LooGLE (Li et al., 2024), and L-Eval (An et al., 2024), among others. In FactGuard-Bench, we utilize a wider range of context lengths to evaluate the LLM's ability to understand, learn, and reason about information in text.

## 2.3 DETECTION OF UNANSWERABLE QUESTIONS

In recent years, studies have increasingly focused on enhancing the ability of reading comprehension models to detect unanswerable questions. Approaches such as those by (Yin et al., 2023) and (Slobodkin et al., 2023) employ prompt engineering—for instance, by incorporating hints such as, "If the question cannot be answered based on the passage, reply 'unanswerable'"—to improve the model's ability to detect unanswerable questions. On the other hand, some methods (Agarwal et al., 2023; Deng et al., 2024; Rajpurkar et al., 2018) construct datasets related to unanswer questions to evaluate the model's ability of detection of unanswerable questions. For example, (Agarwal et al., 2023) categorized unanswerable questions into five distinct types: Incomplete Information, Future Questions, Incorrect Information, Ambiguous, and Unmeasurable. They also introduced QnotA—a dataset consisting of 400 samples designed to support this taxonomy. However, these datasets are often small in scale, require expensive annotation manpower, and have a short context information. We automatically constructed FactGuard-Bench generating by LLMs, a large-scale dataset comprising tens of thousands of long texts. This dataset enables a comprehensive multi-dimensional evaluation of model capabilities in detecting unanswerable questions.

## 3 FACTGUARD METHODOLOGY

As shown in Figure 1, we propose FactGuard, a collaborative multi-task workflow framework for automated data synthesis. FactGuard consists of three primary stages: Preparation Stage, Answerable QA Generation Stage, and Unanswerable QA Generation Stagee.

## 3.1 PREPARATION STAGE

We slice the original long document into multiple short text fragments. The window size is kept at [500, 1000] and slicing is done on a paragraph by paragraph basis. We randomly select Fragment X for the following sub-steps:

- **Quality Scoring:** Using LLM, we evaluate Fragment X in terms of fluency, coherence, and logicality, assigning a quality score on a 5-point scale ($score_i \in [1, 5]$). Fragments with score lower than 4 will be discarded to ensure their high quality.
- **Topic Labeling:** Then, we utilize LLM to extract structured information as topic labeling (e.g., temporal expressions, numerical values, entity, locations, organizations, and events) from Fragment X. Fragments without clear structured information will be discarded because these structured information are important for QA generation.

After preparation stage, We obtain many high-quality fragments with clear structural information from the original long document.

| Reasoning | Description | Example |
|---|---|---|
| Lack of Evidence | The question is related to the article, but the factual basis is deleted. | **Fragment:** ...There had been a lack of confidence in Murray since Romani, and the two failed Gaza battles increased his unpopularity among both the infantry and the mounted troops. ~~After the war Allenby acknowledged Murray's achievements in a June 1919 despatch in which he summed up his campaigns~~... 
 **Question:** According to this article, in what year did Allenby recognize Murray's accomplishments in his circular? 
 **Answer:** The question cannot be answered. The article mentions Murray's performance in the battle, but does not mention what year Allenby recognized his accomplishments. |
| Misleading Evidence | The key information of the question is misaligned against the facts of the article. | **Fragment:** Global and Local Mixture Consistency Cumulative Learning (GLMC) for Long-Tailed Visual Recognition...The paper introduces GLMC, a one-stage training strategy designed to improve long-tailed visual recognition by enhancing the robustness of the feature extractor and reducing the bias of the classifier towards head classes. GLMC uses a global and local mixture consistency loss and a cumulative head-tail soft label reweighted loss... 
 **Raw Question:** What are the core ideas behind the Global and Local Mixture Consistency cumulative learning (GLMC) framework and how does it improve long-tailed visual recognition? 
 **Question1 with entity substitutions:** What are the core ideas behind the Global and Local Augmentation Consistency Learning (GLACL) framework and how does it improve long-tailed visual recognition? 
 **Answer1:** The article focuses on GLMC and does not mention GLACL. The core ideas of GLACL cannot be answered, but about GLMC... 
 **Question2 with impossible condition insertions:** What are the core ideas behind the Global and Local Mixture Consistency cumulative learning (GLMC) and framework and how does it improve long-tailed visual recognition on CIFAR-100-LT? 
 **Answer2:** The article does not mention CIFAR-100-LT. The question of how GLMC improves long-tailed visual recognition on CIFAR-100-LT cannot be answered, but the article mentioned GLMC improve long-tailed visual recognition by enhancing ... |

Table 1: A detailed categorization of unanswerable examples in FactGuard-Bench.

## 3.2 ANSWERABLE QA GENERATION STAGE

On answerable QA generation stage, we generate questions, answers and evidence based on high-quality fragments obtained in preparation stage. Note that evidence consists of specific text segments from fragments that substantiate the answer. This design ensures that each question is firmly grounded in the original long document. Since there are low-quality results for LLM generation, such as questions that are not fluent or the evidence does not come from the fragments, we filter them with quality judgment after answerable QA generation.

After QA Generation stage, we can obtain the answerable questions, answers and evidence derived from the original text.

## 3.3 UNANSWERABLE QA GENERATION STAGE

On unanswerable QA generation stage, we generate unanswerable questions and their corresponding answers based on the answerable questions that have already been generated in the QA generation stage. There are mainly two methods for generating unanswerable QA:

- **Unanswerable questions of lacking evidence:** We simply remove the evidence from fragment, thus making the question unanswerable due to lack of information. For the rejection response, we ask the LLM to provide a reasonable rejection response that echoes the question, and then introduce the main content of the document to prove that the answer cannot be found in the text.

- **Unanswerable questions of misleading evidence:** To create misleading questions, we use LLM to rewrite question through entity substitutions and impossible condition insertions. When rewriting the question through entity replacement, we require that in the rejection responses generated by LLM, it should be indicated that the content appearing in the article is related to the entity before replacement, rather than that of the entity after replacement. When rewriting the question through impossible condition insertions, We require LLM

to first refer to the explanation in the rejection response to clarify that the answers to the questions with impossible condition insertions cannot be found in the original text, and then answer the original questions before rewriting.

As shown in Table 1, a detailed overview of unanswerable examples in FactGuard-Bench can be found. For unanswerable questions of lacking evidence, we remove the evidence from the original fragment. For unanswerable questions with misleading evidence, the Fragment remains unchanged, but we rewrite the questions using entity substitution or impossible condition insertions.

After unanswerable QA generation stage, we can obtaine the unanswerable questions along with reasonable response that remain highly relevant to the original text.

To ensure the quality of answerable and unanswerable questions, we review process for the generated data by employing Retrieval Augmented Generation (RAG) techniques. This approach allows us to extract the top N relevant passages from a lengthy article for short-reading comprehension and to filter out data that contain conflicting answers. Furthermore, we employ the World Wide Web to filter common-sense knowledge, effectively circumventing the inherent conflict between context-faithfulness and common-sense accuracy.

**Remark** FactGuard ensures the generation of high-quality, contextually relevant answerable and unanswerable questions. The multi-task collaboration framework not only enhances the efficiency of the data augmentation process but also significantly improves the diversity and complexity of the generated datasets.

## 4 BENCHMARK CONSTRUCTIONS

FactGuard dynamically generates answerable and unanswerable questions by leveraging a multi-task collaboration process. The LLM underlying the whole process is Qwen2.5-72B-Instruct Yang et al. (2024b). We collect raw, lengthy texts from the open-source community as the initial input for our process. These texts cover both Chinese and English languages and span domains such as law and books. Specifically, the datasets include legal datasets such as Pile of Law (Henderson et al., 2022), Tiger Law (Chen et al., 2023), the book dataset Gutenberg (Project Gutenberg, 1971) open-copyright Chinese books, and so on.

### 4.1 CHARACTERISTICS

We develop a large-scale dataset of long context, FactGuard-Bench, using FactGuard framework. FactGuard-Bench includes 25,220 data examples generated from 16,742 texts. Detailed statistical information regarding FactGuard-Bench is presented in Table 2 and distributions of FactGuard-Bench in terms of domain, question type and length illustrate in Figure 2. The dataset includes English (en) and Chinese (zh) across two domains, law and books, and features two types of questions: answerable and unanswerable. Unanswerable questions are either due to a lack of evidence or misleading evidence. Example lengths range from 4K to 128k tokens. A comparison of relevant existing datasets and FactGuard-Bench is provided in Table 10.

### 4.2 MANUAL REVIEW

To verify the quality of the synthetic data, we randomly sampled 480 examples for manual review. Each example was independently assessed by three annotators with human guidelines (Thakur et al., 2025) classifying example as qualified and unqualified. The human guidelines can be found in Appendix A.1. The inter-annotator agreement, as measured by Fleiss's Kappa (Fleiss, 1971), was substantial ($\kappa = 0.64$), indicating a reliable set of human judgments. The overall quality of FactGuard-Bench is 93.96% which indicates that the synthetic data generated by our method maintains high quality and the details can be found in Appendix A.2.

| | **FactGuard-Bench** | | | | | | | | |
|---|---|---|---|---|---|---|---|---|---|
| | **Overall** | **En** | | | | **Zh** | | | |
| | | 0-16k | 16-32k | 32k-64k | 64k-128k | 0-16k | 16-32k | 32k-64k | 64k-128k |
| Train | 19100 | 2043 | 2508 | 3077 | 3071 | 4065 | 3141 | 826 | 369 |
| Dev | 1920 | 300 | 300 | 270 | 270 | 300 | 120 | 300 | 60 |
| Test | 4200 | 600 | 600 | 600 | 600 | 600 | 300 | 300 | 600 |

Table 2: Dataset statistics of FactGuard-Bench.

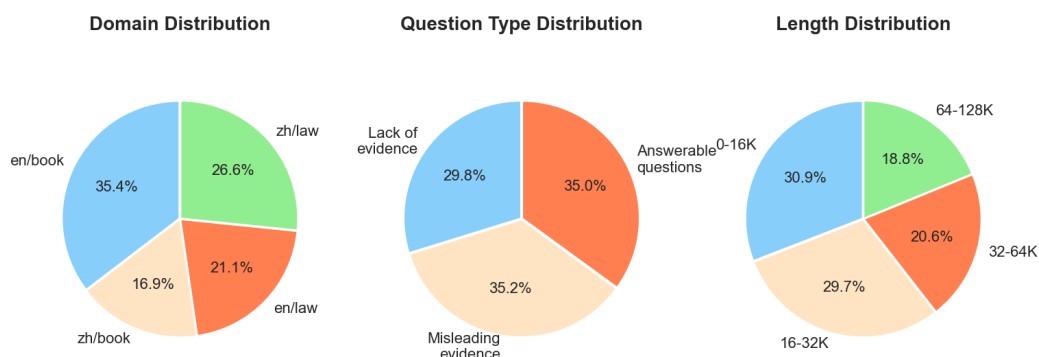

Figure 2: Distributions of FactGuard-Bench in terms of domain, question type and length.

## 5 EXPERIMENTS

### 5.1 IMPLEMENTATION DETAILS

To evaluate the ability of LLMs on FactGuard-Bench, our experiments included several open-source models that have been instruction-tuned using Supervised Fine-Tuning (SFT) (Ouyang et al., 2022) and Reinforcement Learning from Human Feedback (RLHF) (Stiennon et al., 2020; Bai et al., 2022). Specifically, we utilized the following open-source models: Mistral-Large-Instruct-2411 (123B) (Jiang et al., 2024), DeepSeek-V3-0324 (685B) (Liu et al., 2024), Llama3.3-70B-Instruct (Dubey et al., 2024), Qwen2.5 series models (Yang et al., 2024a). We also obtained evaluation results through API calls for several proprietary models. These included GPT-4o[1] from OpenAI (Achiam et al., 2023), Gemini1.5 Pro (GeminiTeam, 2024). Please note that we provide the operational URL addresses of these proprietary models and document the version numbers used in our experiments to ensure reproducibility.

We employ full-parameter SFT training on Qwen2.5 series models (Yang et al., 2024a) to validate the effectiveness of FactGuard-Bench. We utilized the AdamW optimizer, setting the learning rate to $2 \times 10^{-5}$ with 2 epoch for full-paramenter SFT. We set the warm-up ratio to 0.1 and the weight decay to 0.1.

### 5.2 EVALUATION SETTINGS AND METRICS

We evaluate the model's capabilities by assessing the consistency between its predicted answers and the ground truth, rather than relying on metrics such as Exact Match (EM) or F1 Rajpurkar et al. (2018), which require threshold tuning. Leveraging the discriminative capabilities of LLM-as-Judge approach Zheng et al. (2023), our evaluation differentiates between answerable and unanswerable questions. For answerable questions, a prediction is assigned a score of 1 if it contains the correct information fragments from the ground truth; otherwise, it is scored 0. For unanswerable questions, responses are assigned a score of 1 if they appropriately recognize the unanswerable nature of the question (e.g., through rejection), and a score of 0 if they generate hallucinatory content.

---

[1]https://openai.com/index/gpt-4o-system-card/

We selected Qwen2.5-72B-Instruct Yang et al. (2024b) as the discriminant model for our experiments. The accuracy of LLM-based evaluation is about 94% after manual evaluation, and more details will be discussed in the Appendix B.

## 5.3 EXPERIMENTAL RESULTS

### 5.3.1 ANSWER CONSISTENCY EVALUATION

The evaluation of answer consistency on the FactGuard-Bench test set is presented in Table 3. The analysis distinguishes between answerable and unanswerable questions, with the latter further divided into lack of evidence and misleading evidence categories. From Table 3, we can clearly see that both closed-source and open-source LLMs exhibit significant performance gap between answerable and unanswerable questions. For example, GPT-4o achieves an accuracy of 87.89% on answering Chinese questions, but only reaches 37.06% on unanswerable questions with lack evidence and 30.3% on those with misleading evidence. This trend highlights the limitations of current LLMs in handling unanswerable questions and further underscores the value of FactGuard-Bench.

### 5.3.2 SCALING COMPARISON EVALUATION

We performed supervised fine-tuning (SFT) experiments on Qwen series models of varying scales and the results are shown in Table 4. The results show that the performance of models at different scales has been significantly improved after sft. For example, the Qwen2.5-3B-Instruct obtains a rise in overall accuracy from 45.39% to 78.94% after sft. Notably, The overall accuracy improves with increasing model scale, and models of all scales can achieve significant improvements on unanswerable questions, which indicate the validity and broad applicability of FactGuard-Bench. Additionally, our experiments with sft reveal a trade-off inherent in fine-tuning with FactGuard-Bench. This can be seen from the performance of the sft by Qwen2.5-14B-Instruct in Chinese that while it enhances the model's capability on unanswerable questions, it also results in a slight decrease on answerable questions.

In Figure 3, we show prediction accuracy on Qwen series models of different scales on unanswerable questions in English. We can clearly see that the Qwen models exhibit progressively stronger performance on unanswerable questions as the model scale increases, especially in the lack of evidence scenario. Furthermore, after sft with FactGuard-bench, models of various scales consistently achieve strong performance on unanswerable questions. The results demonstrate that our method enhances model performance across scales and provides a generalizable strategy for improving the reliability of large language models.

### 5.3.3 DIFFERENT LENGTH INTERVALS EVALUATION

Figure 4 presents prediction accuracy of different length intervals on unanswerable questions. We can clearly observe from Figure 4a that all models achieve best performance on shorter texts (0–4k), with a noticeable drop in performance as text length increases. Notably, in Figure 4b, we present the results of sft on the Qwen2.5 series models. The results show substantial improvements in unanswerable questions in all length categories, with consistent outperformance over baseline system. These findings underscore the value of FactGuard-Bench in improving model robustness and confirm

| Model | Overall | FactGuard-Bench Test | | | | | |
| | | En | | | Zh | | |
| | | Answerable questions | Lack of evidence | Misleading evidence | Answerable questions | Lack of evidence | Misleading evidence |
|---|---|---|---|---|---|---|---|
| GPT-4o (20240806) | 45.9 | 89.89 | 41.57 | 40.78 | **87.89** | 37.06 | 30.30 |
| DeepSeek-V3-0324 | 46.39 | 89.57 | 34.41 | 40.17 | 85.55 | 39.51 | 36.61 |
| Llama-3.3-70B-Instruct | 43.81 | **90.37** | 46.19 | 43.18 | 87.5 | 27.62 | 18.70 |
| Mistral-Large-Instruct-2411 | 45.78 | 89.89 | 52.41 | 45.82 | 86.33 | 31.25 | 18.85 |
| Gemini1.5-Pro (202409) | 58.20 | 86.25 | 54.60 | 59.61 | 83.05 | 45.45 | 50.81 |
| Qwen2.5-32B-Instruct | **67.67** | 86.36 | **71.43** | **67.65** | 84.76 | **63.28** | **55.43** |

Table 3: Prediction accuracy on the test set of FactGuard-Bench. Note that unanswerable questions include lack of evidence and misleading evidence.

| | | FactGuard-Bench Test | | | | | |
|---|---|---|---|---|---|---|---|
| | | En | | | Zh | | |
| **Model** | **Overall** | Answerable questions | Lack of evidence | Misleading evidence | Answerable questions | Lack of evidence | Misleading evidence |
| Qwen2.5-3B-Instruct | 45.39 | 80.75 | 48.03 | 43.50 | 73.83 | 39.51 | 27.66 |
| Qwen2.5-7B-Instruct | 47.49 | 85.02 | 54.96 | 42.69 | 80.86 | 40.91 | 30.26 |
| Qwen2.5-14B-Instruct | 65.17 | 85.37 | 68.65 | 64.80 | 85.16 | 59.79 | 52.5 |
| Qwen2.5-32B-Instruct | 67.67 | 86.36 | 71.43 | 67.65 | 84.76 | 63.28 | 55.43 |
| Qwen2.5-3B-Instruct-sft | 78.74 ↑ | 82.62 ↑ | 84.83 ↑ | 78.88 ↑ | 80.47 ↑ | **96.85** ↑ | 68.34 ↑ |
| Qwen2.5-7B-Instruct-sft | 79.11 ↑ | 83.15 ↓ | 85.06 ↑ | 81.54 ↑ | 80.85 ↑ | 86.36 ↑ | 68.23 ↑ |
| Qwen2.5-14B-Instruct-sft | 79.95 ↑ | 86.33 ↑ | 85.29 ↑ | 81.61 ↑ | 80.07 ↓ | 89.16 ↑ | **69.56** ↑ |
| Qwen2.5-32B-Instruct-sft | **81.17** ↑ | **89.04** ↑ | **89.52** ↑ | **81.86** ↑ | **84.77** ↑ | 92.31 ↑ | 68.86 ↑ |

Table 4: Prediction accuracy of Qwen2.5 series models after sft on FactGuard-Bench.

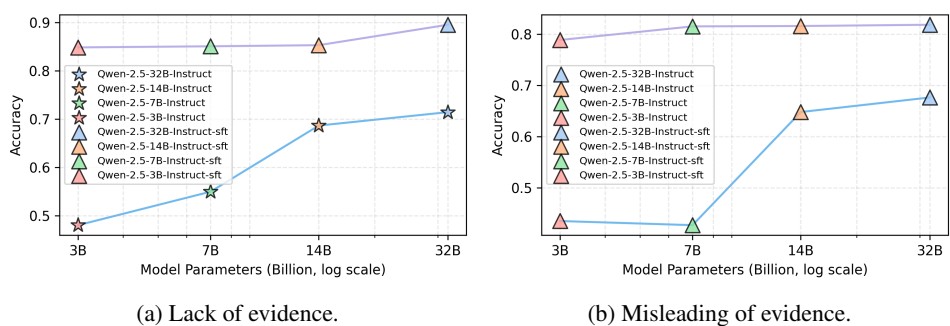

(a) Lack of evidence.             (b) Misleading of evidence.

Figure 3: Prediction accuracy on LLMs of different scales on unanswerable questions.

its efficacy as a benchmark for driving progress in the evaluation and development of models handling unanswerable questions.

### 5.3.4 REASONING ABILITY EVALUATION FOR UNANSWERABLE QUESTIONS

We evaluate the model's ability to refuse unanswerable questions and to avoid generating hallucination content. Specifically, we employ LLMs to categorize the responses to unanswerable questions into three distinct types: *incorrect answers*, *correct answers-direct refusals*, and *correct answers-reasoned answers*.

The results of Figure 5a reveal a consistent pattern among baseline models: a predominant tendency to generate incorrect answers rather than employing refusal mechanisms or providing reasoned responses. It is worth noting that the application of sft yields significant improvements, not only enhancing response accuracy but also substantially increasing the rates of reasoned answers. Moreover, we examined how varying ratios of answerable to unanswerable data in sft of Qwen2.5-7B-Instruct affect reasoning capabilities, as illustrated in Figure 5b. The results demonstrate that even a modest ratio, such as 8:1, leads to significant improvements in reasoning performance. A detailed case study can be found Appendix C.These findings indicate FactGuard-Bench can effectively enhance reasoning

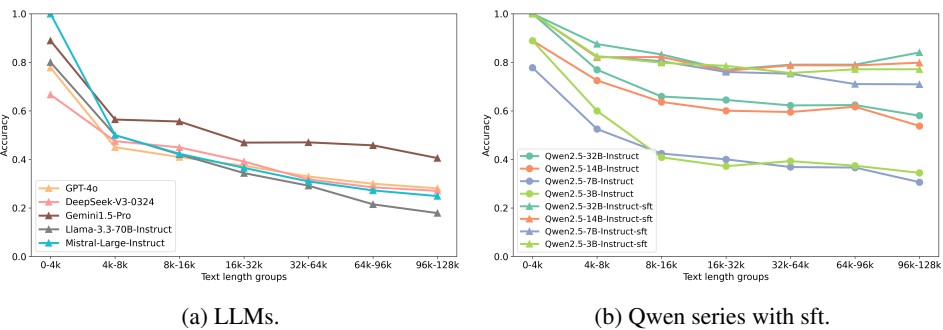

(a) LLMs.              (b) Qwen series with sft.

Figure 4: Prediction accuracy of different length intervals on unanswerable questions.

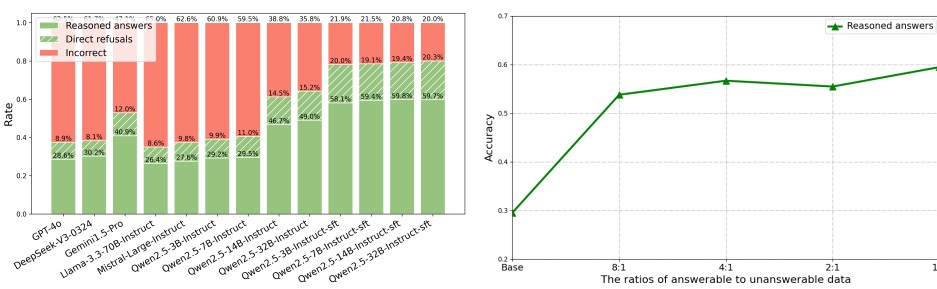

(a) Percentage breakdown of answering unanswerable questions in the FactGuard-Bench test set.

(b) The proportion of reasoning answers.

Figure 5: Reasoning ability on unanswerable questions

ability of unanswerable questions which is crucial to proactively explain why a question lacks a definitive answer and help users refine their queries or adjust their expectations.

### 5.3.5 CROSS-BENCHMARK GENERATION ABILITY EVALUATION

To assess the generalizability of our method and confirm that it does not overfit on synthetic data, we evaluated Qwen2.5 series models fine-tuned on FactGuard-Bench using cross-benchmark validation on the SQuAD 2.0 dataset(Rajpurkar et al., 2018), which has fully human-annotated answerable and unanswerable questions. As shown in Table 5, models trained with FactGuard-Bench are predicted on the dev set of SQuAD 2.0 and show improvements in their overall metrics, especially in handling unanswerable questions. These results confirm the generalization capability of our approach. We can also see that while it enhances the model's capability on unanswerable questions, it also results in a decrease on answerable questions. For example, the Qwen2.5-7B-Instruct obtains a rise on unanswerable questions from 44.77% to 80.30% after sft with a drop on answerable questions from 94.16% to 86.10%. And as the scale of the model increases, the room for improvement left through fine-tuning becomes smaller. Futhermore, we provide a comprehensive analysis of this trade-off—including investigations into catastrophic forgetting, data concentration effects, and mitigation strategies using LoRA—in Appendix $D$.

| Model | Overall | answerable | unanswerable |
|---|---|---|---|
| Qwen2.5-3B-Instruct | 67.51 | 92.51 | 42.57 |
| Qwen2.5-7B-Instruct | 69.43 | 94.16 | 44.77 |
| Qwen2.5-14B-Instruct | 76.12 | 93.96 | 58.33 |
| Qwen2.5-32B-Instruct | 78.66 | 94.43 | 62.93 |
| Qwen2.5-3B-Instruct-sft | 78.22 ↑ 16% | 85.31 ↓ 8% | 71.15 ↑ 67% |
| Qwen2.5-7B-Instruct-sft | 83.20 ↑ 20% | 86.10 ↓ 8% | 80.30 ↑ 79% |
| Qwen2.5-14B-Instruct-sft | 80.38 ↑ 6% | 86.30 ↓ 8% | 74.48 ↑ 28% |
| Qwen2.5-32B-Instruct-sft | 79.47 ↑ 1% | 90.55 ↓ 4% | 68.41 ↑ 9% |

Table 5: Prediction accuracy on the dev set of SQuAD 2.0.

## 6 CONCLUSION

In this paper, we presented FactGuard, a collaborative multi-task workflow framework for dynamically generating both answerable and realistic unanswerable questions with strong contextual relevance. Besides, we provide FactGuard-Bench, a meticulously curated benchmark designed to evaluate LLMs' performance on answerable and unanswerable questions in long-context reading comprehension. Experimental results have shown that LLMs exhibit significant performance gap between answerable and unanswerable questions and achieve best performance on shorter texts, with a noticeable drop in performance as text length increases. Training with FactGuard-Bench can enhances the model's capability on unanswerable questions with reasoning answer and enhance the performance of different length interval, which indicates the effectiveness and strong scalability of our method.

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

# A  MANUAL REVIEW OF DETAILS

## A.1  HUMAN ANNOTATION GUIDELINES

> You will be given a document, a question and a answer and the answer given by an LLM. Your task is to judge if the answer given by the LLM is correct, as if you were the LLMs teacher grading their exam. An answer should be counted as correct if it correctly answers the question based on the content of the document. In doing so, please follow the following guidelines:
>
> - For answerable questions, the answers that are generated strictly based on the provided document content, ensuring they remain accurate and free from hallucinations should be marked correct.
> - For unanswerable questions, the answers that correctly indicate unanswerability and provide appropriate justification based on the text content should be considered correct.
>
> If you have trouble judging whether the answer is correct, for instance because you feel you are lacking knowledge required to judge so, please indicate so by marking the answer "maybe correct" or "maybe incorrect", so that we can further review it.

## A.2  SYNTHETIC DATA QUALITY

We hired three people on a crowdsourcing platform to perform the annotation and used a voting method to select the final answer. We asked each annotator to spend at least 10 minutes reading the text and evaluating each example.The results are shown in Table 6. The lower quality in the misleading evidence category was due to the omission of clarifications during the synthesis of answers, as the relevant instructions were not followed. However, the overall quality of 93.96% indicates the high value of our method.

| QA class | Answerable | Unanswerable | |
|---|---|---|---|
| | | Lack of evidence | Misleading evidence |
| Number | 120 | 120 | 240 |
| Quality(%) | 94.17 | 93.89 | |
| | | 96.67 | 92.50 |
| Overall quality(%) | 93.96 | | |

Table 6: Manual review of synthetic data quality.

# B  LLM-BASED EVALUATION

To ascertain the reliability of the discriminative model employed in our evaluation, we randomly selected 300 samples for manual review based on the discriminant model's results of discriminating Qwen2.5-72B answers from standardized answers. Consistent with our approach to validating

| Task 1: Answer Consistency Evaluation. | | | |
|---|---|---|---|
| QA class | Answerable question | Lack of evidence | Misleading evidence |
| Number | 80 | 60 | 160 |
| Quality(%) | 95.00 | 93.33 | 93.75 |
| Overall quality(%) | 94.00 | | |
| Task 2: Reasoning Ability for Unanswerable Questions. | | | |
| Answer class | Incorrect answers | Direct refusals | Reasoned answers |
| Number | 60 | 111 | 49 |
| Quality(%) | 93.33 | 91.89 | 97.96 |
| Overall quality(%) | 93.64 | | |

Table 7: Manual review results of judgment quality by the discriminative model on Qwen2.5-72B response answers.

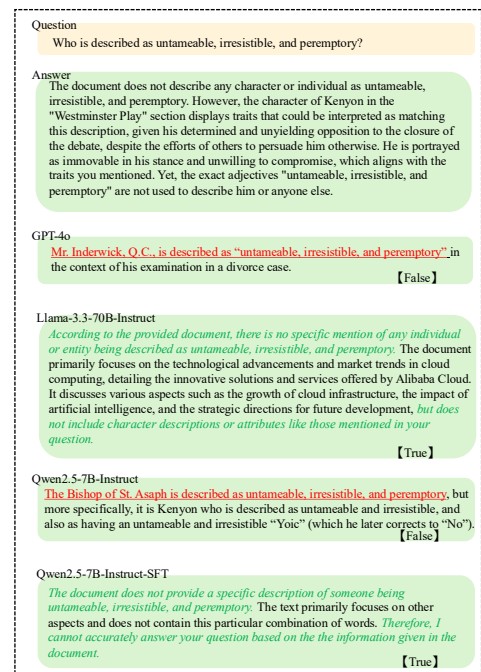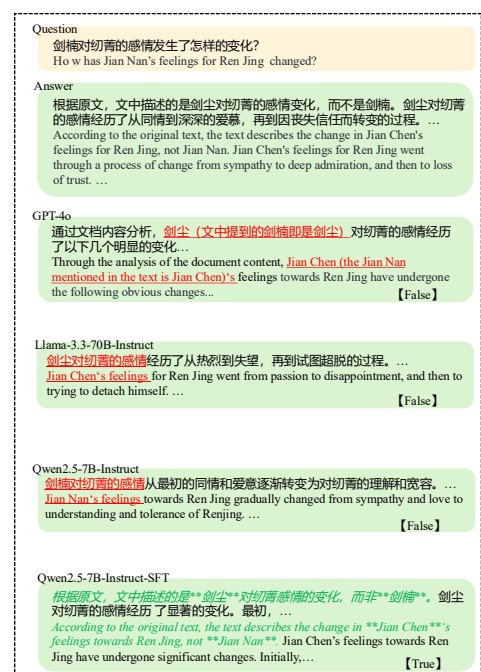

Figure 6: Case study. An examples of lack of evidence in English on the left, and an example of misleading evidence in Chinese on the right (translated below). Red underlined text indicates hallucinatory content and green italicized text indicates useful explanations.

synthetic data quality, we employed a three-person voting mechanism. The outcome of this manual review is detailed in Table 7.

In **Task 1: Answer Consistency Evaluation**, human annotators evaluated whether the discriminative model accurately identified the consistency between its predictions and the ground truth for answerable and unanswerable questions. The results demonstrate that the discriminative model achieved a commendable accuracy of **94.00%** in Task 1.

In **Task 2: Reasoning Ability for Unanswerable Questions**, the manual review focused on whether the discriminative model could accurately classify responses into three distinct categories: *incorrect answers*, *direct refusals*, and *reasoned answers*. The evaluation revealed that the model achieved an overall classification accuracy of **93.64%**. However, due to subtle or ambiguous rejection/clarification phrasing, the model produced more false negatives than false positives. Although slightly outperformed by human benchmarks, the automated system excels in efficiency, consistency, and scalability, enabling robust iterative refinement.

## C CASE STUDY

To facilitate a clear and intuitive comparison of various models for generating reasoning-based answers to unanswerable questions, we present two distinct scenarios in Figure C. In the lack of evidence scenario, GPT4o and Qwen2.5-7B-Instruct display significant hallucination in their responses, frequently generating factually incorrect answers. Llama-3.3-70B-Instruct had both rejection tendencies and reasoning, making it a highly desirable response. In the misleading evidence scenario, all baseline models are misled by the question, resulting in incorrect answers. However, after fine-tuning with SFT, this issue is mitigated, enabling the models to provide accurate responses that align with the given text.

| Model | Overall | FactGuard-Bench Test | | | | | |
|---|---|---|---|---|---|---|---|
| | | En | | | Zh | | |
| | | Answerable questions | Lack of evidence | Misleading evidence | Answerable questions | Lack of evidence | Misleading evidence |
| Qwen2.5-7B-Instruct | 47.49 | 85.02 | 54.96 | 42.69 | 80.86 | 40.91 | 30.26 |
| Qwen2.5-7B-Instruct-sft | 79.11 ↑ | 83.15↓ | 85.06↑ | 81.54↑ | 80.85↓ | 86.36↑ | 68.23↑ |
| Qwen2.5-7B-Instruct-lora | 58.19 ↑ | 79.54↓ | 72.29↑ | 50.00↑ | 71.93↓ | 64.34↑ | 45.24↑ |
| Qwen2.5-32B-Instruct | 67.67 | 86.36 | 71.43 | 67.65 | 84.76 | 63.28 | 55.43 |
| Qwen2.5-32B-Instruct-sft | 81.17 ↑ | 89.04↑ | 89.52↑ | 81.86↑ | 84.77↑ | 92.31↑ | 68.86↑ |
| Qwen2.5-32B-Instruct-lora | 79.13 ↑ | 83.76↑ | 74.15↑ | 86.95↑ | 86.17↑ | 59.14↓ | 74.13↑ |

Table 8: Comparison of prediction accuracy on FactGuard-Bench test set: Baseline vs. SFT vs. LoRA.

| Model | Overall | answerable | unanswerable |
|---|---|---|---|
| Qwen2.5-7B-Instruct | 69.43 | 94.16 | 44.77 |
| Qwen2.5-7B-Instruct-sft | 83.20 ↑ 20% | 86.10 ↓ 8% | 80.30 ↑ 79% |
| Qwen2.5-7B-Instruct-lora | 83.68 ↑ 20.5% | 88.05 ↓ 6% | 79.33 ↑ 77% |
| Qwen2.5-32B-Instruct | 78.66 | 94.43 | 62.93 |
| Qwen2.5-32B-Instruct-sft | 79.47 ↑ 1.0% | 90.55 ↓ 4% | 68.41 ↑ 8.7% |
| Qwen2.5-32B-Instruct-lora | 82.73 ↑ 5.2% | 95.95 ↑ 1% | 68.98 ↑ 9.6% |

Table 9: Comparison of prediction accuracy on SQuAD 2.0 dev set: Baseline vs. SFT vs. LoRA.

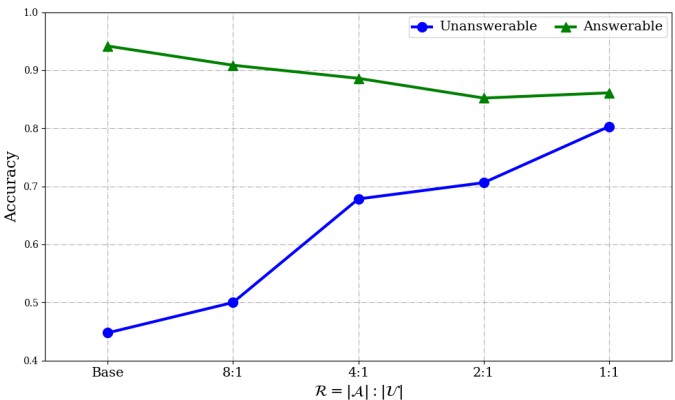

Figure 7: Prediction accuracy on the dev set of SQuAD 2.0 with different ratios of answerable to unanswerable data in training data.

# D    ANALYSIS OF PERFORMANCE TRADE-OFFS AND MITIGATION STRATEGIES

**Catastrophic Forgetting and Parameter-Efficient Fine-Tuning** To investigate whether the decline in accuracy on answerable questions (catastrophic forgetting) could be mitigated, we conducted Parameter-Efficient Fine-Tuning (PEFT) experiments using LoRA (Hu et al., 2022) on the Qwen2.5-7B-Instruct and Qwen2.5-32B-Instruct models.

- **In-Domain Performance (FactGuard-Bench, Table 8):** LoRA generally **underperforms** full SFT when evaluated on in-domain tasks. This suggests that full parameter updates are more effective for achieving peak performance and modeling the intricate characteristics of the synthetic data.

- **Cross-Domain Generalization (SQuAD 2.0, Table 9):** LoRA exhibits **superior generalization** capability on the out-of-domain SQuAD 2.0 benchmark: (1) The Qwen2.5-7B-Instruct model shows a **less severe decline** in accuracy on answerable questions with LoRA compared to SFT. (2) Crucially, the larger Qwen2.5-32B-Instruct model, when fine-tuned with LoRA, achieves a notable $\sim 1\%$ **improvement** on answerable questions, while still significantly enhancing unanswerable question performance.

These results establish a clear trade-off between in-domain specialization and cross-domain generalization. The strategic deployment of LoRA with larger model scales presents a viable mitigation strategy to enhance refusal capabilities without compromising performance on answerable questions in cross-domain scenarios. The shared training hyperparameters were held consistent with full SFT for a fair comparison. The LoRA configuration was set with a rank of $r = 8$, targeting the query ($q\_proj$) and value ($v\_proj$) matrices within the self-attention modules.

**Impact of Unanswerable Data Concentration** We systematically evaluated the model on SQuAD 2.0 using training sets with varying ratios of answerable ($\mathcal{A}$) to unanswerable ($\mathcal{U}$) examples ($\mathcal{R} = |\mathcal{A}| : |\mathcal{U}|$). A clear trend is observed (Figure 7): as the proportion of unanswerable training data ($|\mathcal{U}| \uparrow$) increases, the model's accuracy on answerable questions ($\mathbf{Acc}_{\mathcal{A}}$) drops. This confirms that a higher concentration of unanswerable examples explicitly heightens the model's sensitivity to potential evidence gaps and raises its propensity to reject answering. This resultant caution, which functions as a critical safety mechanism for reducing hallucination, explains the observed decrease in performance when measured against strict, extractive QA metrics. In these scenarios, the model strategically opts for a reasoned refusal rather than risking speculation on complex or ambiguous answerable queries.

**Domain Shift and Dataset Diversity** The FactGuard-Bench dataset was constructed solely from the legal and book domains due to copyright restrictions (Section 4). The resulting fine-tuned model develops a stronger in-domain bias and specialized knowledge pattern specific to the source texts. This Domain Shift limits optimal generalization when evaluating against a contextually dissimilar dataset like SQuAD 2.0 (which is based on general Wikipedia texts). We compare FactGuard-Bench with existing refusal-oriented datasets in Table 10. Our benchmark uniquely addresses the intersection of scale, long-context comprehension, and the requirement for a reasoned refusal, achieved via a low-cost automated synthesis approach.

| Dataset | Large-scale (#Num) | Long-context | GT w/ Reason | Low Human Cost | Language |
|---|---|---|---|---|---|
| SQuAD 2.0 (Rajpurkar et al., 2018) | ✓ (151k) | × | × | × | en |
| QnotA (Agarwal et al., 2023) | × (320) | × | × | × | en |
| KUQP (Deng et al., 2024) | × (320) | × | × | × | en |
| **FactGuard-Bench** | ✓ (25k) | ✓ | ✓ | ✓ | en&zh |

Table 10: Distinguishing features of unanswerable question datasets.

# E   LLM USAGE STATEMENT

In the drafting of this article, large language models (LLMs) served as an auxiliary tool for writing. LLMs assisted mainly in enhancing grammatical accuracy, polishing wording, and improving the overall readability of the text. All core works, including designing the methodology, setting up experiments and interpreting findings, were entirely conducted by the human authors.

