# OpenReview forum: "FactGuard: Detecting Unanswerable Questions in Long-Context Texts for Reliable LLM Responses"
_ICLR.cc/2026/Conference — Submitted to ICLR 2026_

### Official Review · Reviewer_3jaj · 2025-10-30

**Soundness:** 2
**Presentation:** 2
**Contribution:** 3
**Rating:** 4
**Confidence:** 3

**Summary:**

This paper addresses the problem of recognizing unanswerable questions when querying large language models (LLMs). The authors propose a workflow, FactGuard, for generating question–answer (QA) pairs with defined answerability/unanswerability from documents such as books and legal documents. Using this workflow, they construct the FactGuard-Bench dataset, which contains approximately 25,000 answerable and unanswerable questions with varying context lengths. The authors evaluate both open-source and commercial LLMs on this dataset and find notable performance gaps between answerable and unanswerable questions. They further analyze the impact of supervised fine-tuning with FactGuard-Bench on model performance.

**Strengths:**

- The problem addressed in the paper is well-motivated and interesting.
- The dataset is substantial in size and exhibits diverse context lengths. It also maintains a balanced distribution between answerable and unanswerable questions.
- The evaluation is comprehensive and considers multiple aspects of the problem.

**Weaknesses:**

- A large portion of the proposed framework and evaluation pipeline relies on LLMs. For instance, text quality scoring, topic labeling, question–answer generation, and even the evaluation of LLM-generated answers are all performed by LLMs. This extensive use of LLM-based components raises concerns about the reliability and objectivity of the resulting dataset. The authors should clarify why alternative, non-LLM approaches were not considered for these steps or provide justification for the exclusive reliance on LLMs.
- Some important methodological details are missing. In particular, Section 3 lacks sufficient clarity regarding how the steps were implemented. For example, the specific prompts used, input–output formats, and parameter settings. Providing these details would greatly help understand the method better

Minor comments: lines 80 and 221 have redundant words.

**Questions:**

- What is the text associated with each question-answer pair in the dataset? The entire document or only the fragment that was used to generate the question?
- How do you prompt the LLMs to answer FactGuard questions? Do you provide the text chunk in the prompt as well? If yes, how? Do you call the LLM multiple times and report an average accuracy or only one time? Is it greedy decoding?
- How do you explain the decrease in accuracy for answerable questions when performing SFT in section 5.3.5?
- Could you explain a bit more about lines 225 to 230? This part was confusing for me.

---

> ### Author Response · Authors · 2025-11-21
>
> We sincerely appreciate your detailed comments, which have helped us better clarify potentially ambiguous aspects of the paper and further enhance the quality of our work.
>
> # Responses to Weaknesses and Questions
>
> ## **W1: LLMs as both generators and evaluators**
> ### **A. Necessity of LLM Scaling:**
> The primary reason for relying on LLM-based approaches is the scale and complexity of the task. Manually annotating the entire dataset—which contains 25,220 question-answer pairs, including **reasoned rejections**, derived from documents with up to 128K tokens—would be **prohibitively expensive and time-consuming**. LLM synthesis is the feasible path to construct a benchmark of this scale and complexity.
>
> ### **B. Mechanisms for Objectivity and Reliability:**
> Our framework does not simply rely on raw LLM output. We have incorporated several key mechanisms to enhance objectivity and reliability:
> - RAG-Grounded Generation: Answerable questions are generated based on evidence retrieved by a RAG system from the source documents, tethering the content to the provided text rather than the model's internal knowledge.
> - World Knowledge Filtering: We actively filter out questions that can be answered using common world knowledge, ensuring the questions are truly context-dependent.
> - Human-in-the-Loop Quality Control: As presented in the paper, we performed rigorous human validation on a sample of the LLM-generated data. The high agreement between human annotators and the LLM-as-a-Judge results demonstrates the reliability of our automated evaluation.
>
> ## **W2: Methodological details**
> **Open Source:** Considering that the numerous prompts in the workflow would take up too much space, the corresponding prompts and data formats will be made publicly available in a subsequent open-source release of the codebase.
>
> **Redundant words:** Thank you for catching these errors. We have corrected them.
>
> ## **Q1: The text associated with each question-answer pair in the datase**
> The text associated with each question-answer pair in the dataset is the entire document and we generate each question-answer (QA) pair from an individual fragment of original longer document.
>
> ## **Q2: How do you prompt the LLMs to answer FactGuard questions?**
> The prompt follows a structured template, such as: "Document: [Related fragment]. Please answer the question based on the document: [Question]". This should only be invoked once and is not subject to greedy decoding, because we have observed that answer generation for short passages is simpler and more stable than for long passages.
>
> ## **Q3: The performance on answerable questions decreases**
> The observed decrease in answerable accuracy during SFT is an expected capacity trade-off. We provide the following explanation and mitigation:
> ### **A. Mitigation via LoRA:**
> The LoRA experiment demonstrates that the performance drop is manageable. The detailed result can be found in Table 8 and 9. The LoRA results for Qwen2.5-32B demonstrate that PEFT successfully enhances the model's ability to handle unanswerable questions while **simultaneously improving performance on answerable questions** in cross-domain scenarios (94.43% $\to$ 95.95%).
>
> ### **B. Contributing Factors:**
> **Learned Rejection Tendency:** The high concentration of unanswerable examples in the dataset explicitly heightens the model's caution.
>
> **Domain Shift:** The specialization on Legal and Book domains limits immediate generalization to the external Wikipedia-based SQuAD 2.0 text distribution.
>
> **The details of explaining the performance degradation on answerable questions can be found in Appendix D and in our response to Reviewer HKQn (W1&Q1).**
>
> ## **Q4: Clarification on lines 225-230**
> We generate each question-answer (QA) pair from an individual fragment of a longer document. And each data instance in FactGuard-Bench consists of a long document and its corresponding question-answer pair. For answerable questions, other fragments may contain alternative answers to the same question; for unanswerable questions, other fragments might contain evidence that could address the query.
>
> To ensure data quality, we employ a retrieval-augmented generation (RAG) approach for each data instance, retrieving the N most relevant fragments from the remainder of the text (excluding the fragment used for QA pair generation). If any of these retrieved fragments can answer the question, the entire data instance is discarded.
>
> Additionally, we leverage the World Wide Web to prune instances of common knowledge. This ensures that questions cannot be answered from the model's internal knowledge alone, forcing both questions and answers to be strictly grounded in the provided text.

---

### Official Review · Reviewer_YG22 · 2025-10-30

**Soundness:** 3
**Presentation:** 2
**Contribution:** 2
**Rating:** 4
**Confidence:** 4

**Summary:**

This paper proposes a framework called FactGuard for generating both answerable and unanswerable questions. The unanswerable QA pairs are created by either modifying the fragment (used as context) or modifying the question, resulting in two types of unanswerable questions: lacking evidence and misleading evidence. The authors then apply their framework to construct the FactGuard-Bench benchmark, which contains 8,829 answerable and 16,391 unanswerable questions. Experimental results reveal a performance gap between answerable and unanswerable questions, with models struggling particularly on the latter. The authors also show that model performance improves after fine-tuning on their dataset, with models achieving their best results on shorter texts.

**Strengths:**

- Fine-tuning the models on their dataset improves performance on unanswerable questions across different model sizes, as well as on the SQuAD 2.0 dataset.
- Their dataset is interesting because it reveals the weaknesses of existing models in answering unanswerable questions (Table 3).
- They conduct experiments on a wide range of models, making the evaluation quite comprehensive.

**Weaknesses:**

- The manual review process for the dataset is not clearly explained (see questions). In addition, the inter-annotator agreement appears relatively low (0.64). There is also no human baseline reported for the task; based on my understanding, the overall quality score does not represent a human baseline.
- Regarding the cross-benchmark generation ability evaluation, the authors only evaluate on one dataset, SQuAD 2.0, which is relatively old (2018). This does not sufficiently demonstrate the effectiveness of their dataset.
- When fine-tuning on their dataset, the performance on answerable questions decreases, yet the authors do not provide any explanation for this (see Table 5).

**Questions:**

-The terms significant or significantly are used frequently, but no p-values are reported.
- Section 5.3.5: Are there any other datasets that could be used for evaluation here, since SQuAD 2.0 is quite old?
- How do you prompt models to solve the task? What is your approach to task prompting?
- Section 4.2 Manual Review: in 480 examples here, how many are answerable and how many are unanswerrable questions.
- Section 4.2 Manual Review: In the human guidelines in Appendix A.1, you also include the labels maybe correct and maybe incorrect. How are these labels used when calculating the inter-annotator agreement?
- Section 4.2 and Appendix A.2: You mention that annotators are asked to “read the text and evaluate each example,” so is the task not to find the answer to the question, but rather to check whether the existing answer is correct?



Comments/Typos/Suggestions:

- It would be useful to provide a comparison table between existing datasets and your proposed dataset, highlighting the features that are unique to your dataset.

- Many of the papers in the references were published at conferences, but the citations refer only to their arXiv versions. For example https://aclanthology.org/2024.acl-long.506/ and https://aclanthology.org/2023.emnlp-main.220/
- Line 221: repeat “Lacking evidence”
- Line 080 or 081: “we achieved achieves an”
- Line 247: Footnote misuse

---

> ### Author Response · Authors · 2025-11-21
>
> Thank you for the valuable feedback. It has helped us further improve the flaws in the paper.
> The critical points raised regarding transparency and the performance trade-off have been addressed with detailed clarifications and new experimental evidence (LoRA and KUQA results). We hope that these additions warrant increasing the confidence in our submission.
>
> # Responses to Weaknesses and Questions
>
> ## **W1&Q3Q4Q5: Manual Review**
> In our human evaluation, annotators assessed **120 answerable** and **360 unanswerable questions** (detailed in Appendix A.2, Table 6), focusing on the factual consistency and reasoning quality of the answers. Both the "maybe correct" and "maybe incorrect" judgments are categorized under the "incorrect" label.
>
> The inter-annotator consistency, measured by **Fleiss' Kappa ($\kappa$)**[1], reached substantial agreement ($\kappa = 0.64$), demonstrating the reliability of our human judgments.
>
> For context, similar annotation efforts have reported comparable reliability, such as in paper [2], where the overall agreement was $\kappa = 0.598$ (moderate) and answerability-specific agreement was **$\kappa = 0.679$ (substantial)**. Our result is thus highly competitive and supports the quality of our evaluations.
>
> ## **W2&Q1: Cross-benchmark Ability Evaluation**
>
> FactGuard-Bench is designed not only to address answer answerable questions based on long contexts but also to provide well-reasoned explanations for unanswerable ones. While several recent datasets focus on detecting answerability and lack the corresponding reference answers, SQuAD 2.0 includes both answerable questions with ground-truth answers and unanswerable questions, making it suitable for our scenario.
>
> To further strengthen the cross-domain evaluation, we performed a supplemental evaluation on the fine-tuned Qwen2.5-7B-Instruct model using an incomplete-type questions task from the recent KUQA dataset [3], which involves determining the answerability of a question given a passage:
> | Model | Overall Acc. | Answerable Acc. | Unanswerable Acc. |
> | :--- | :---: | :---: | :---: |
> | Qwen2.5-7B-Instruct (Baseline) | 78.75 | 97.5 | 60 |
> | Qwen2.5-7B-Instruct-sft | $\mathbf{87.5} \uparrow$ | $\text{80} \downarrow$ | $\mathbf{95} \uparrow$ |
>
> The results indicate that our method enhances performance in handling unanswerable questions within cross-domain settings, which is consistent with the findings on SQuAD 2.0. It is important to note that the KUQA dataset is small, with only 320 samples. Therefore, this paper will first present the experimental results of SQuAD 2.0.
>
>
> ## **W3: The performance on answerable questions decreases**
>
> **A. Mitigation via LoRA (Catastrophic Forgetting):**
> Our new PEFT results using LoRA decisively mitigate the concern that SFT causes irrecoverable degradation (Table 8 and Table 9).
> The LoRA results for Qwen2.5-32B demonstrate that PEFT successfully enhances the model's ability to handle unanswerable questions while **simultaneously improving performance on answerable questions** in cross-domain scenarios (94.43% $\to$ 95.95%).
>
> **B. Cause: Increased Rejection Propensity \& Domain Shift:**
> The decline in SFT is primarily due to the learned **Increased Rejection Tendency** (Figure 7) and the effect of **Domain Shift** (training only on Legal and Book texts).
>
> **For a more detailed explanation of the performance degradation associated with answerable questions, please see Appendix D and Responses of Reviewer HKQn W1&Q1.**
>
> ## **Response to Comments and Missing Details**
> **Comparison Table**
> A comprehensive comparison table (Table 10 in the Appendix) has been added, highlighting FactGuard-Bench's unique features ($\mathbf{Long-context}$, $\mathbf{GT w/ Reason}$, $\mathbf{Low\ Human\ Cost}$) against prior work.
>
> **About Prompt**
> We commit to providing full LLM prompting details in the open-source code.
>
> **Typos, References, Footnote, etc.**
> We have made corrections. If anything is missing, please point it out. Thank you again.
>
> ## **References**
>
> [1] Joseph L Fleiss. Measuring nominal scale agreement among many raters. Psychological bulletin, 1971.
>
> [2] Li H, et al. Ditch the gold standard: Re-evaluating conversational question answering, Proceedings of the 60th Annual Meeting of the Association for Computational Linguistics (Volume 1: Long Papers), 2022.
>
> [3] Yang Deng, et al. Don’t just say “i don’t know”! self-aligning large language models for responding to unknown questions with explanations. Association for Computational Linguistics, 2024.

---

### Official Review · Reviewer_HKQn · 2025-11-01

**Soundness:** 3
**Presentation:** 3
**Contribution:** 4
**Rating:** 6
**Confidence:** 4

**Summary:**

This paper addresses the critical and timely problem of large language models (LLMs) hallucinating answers to unanswerable questions, with a specific and novel focus on long-context scenarios (up to 128K tokens). The authors make two primary contributions to the community. First, they propose "FactGuard," an innovative and scalable multi-task workflow for automatically synthesizing a large dataset of both answerable and unanswerable question-answer pairs from raw documents. Second, using this workflow, they construct and release "FactGuard-Bench," a new, large-scale benchmark comprising over 25,000 examples designed to evaluate and enhance the ability of LLMs to handle unanswerable queries in long texts. The paper empirically demonstrates that current state-of-the-art LLMs struggle significantly with this task and shows that fine-tuning on FactGuard-Bench can substantially improve a model's ability to correctly identify and provide reasoned refusals for unanswerable questions.

**Strengths:**

- **Pioneering and Valuable Benchmark:** The most significant contribution of this work is FactGuard-Bench. To my knowledge, it is the first large-scale benchmark specifically designed to test unanswerability detection within the challenging domain of very long contexts. As the community pushes the boundaries of context length, this benchmark provides an essential and much-needed tool for evaluating the reliability of these powerful models, moving beyond simple "needle-in-a-haystack" tests.

- **Systematic and Scalable Data Generation:** The FactGuard workflow is a well-designed methodology for data synthesis. The systematic creation of unanswerable questions through "evidence deletion" and, more impressively, "misleading evidence" (entity substitution, impossible conditions) produces challenging examples that test for deep contextual understanding rather than superficial keyword matching. This automated approach ensures scalability.

- **Comprehensive Empirical Validation:** The authors conduct a thorough evaluation across nine prominent LLMs, including GPT-4o, convincingly demonstrating that the inability to handle unanswerable questions in long contexts is a widespread and significant problem. This robustly motivates the need for their work.

**Weaknesses:**

- **Significant Performance Trade-off:** The paper commendably reports a critical weakness: fine-tuning on FactGuard-Bench, while improving performance on unanswerable questions, leads to a notable degradation in accuracy on answerable questions. This suggests the model may become overly cautious, potentially harming its utility in practical applications where users expect correct answers to valid queries. This trade-off is a major barrier to the direct deployment of this fine-tuning method.

- **Narrow Definition of Unanswerability:** The benchmark's scope is limited to contextual fidelity—whether an answer can be found within the provided text. This is a narrow slice of the unanswerability problem. More comprehensive benchmarks like AbstentionBench cover a wider range of epistemic uncertainties, such as questions with false premises, subjective queries, or those requiring knowledge beyond the model's cutoff date. FactGuard-Bench effectively evaluates a model as a "careful reader" but not necessarily as a "knowledgeable and cautious agent."

- **Reliance on a Single Methodological Paradigm:** The paper's solution relies exclusively on supervised fine-tuning (SFT). However, the field is rapidly exploring alternative, potentially more efficient paradigms. These include methods that probe a model's intrinsic self-awareness of its knowledge boundaries (e.g., by analyzing hidden states or reasoning trajectories) or employ mechanistic interventions (e.g., activation steering). These approaches may offer a path to improving refusal capabilities without the performance trade-offs associated with SFT.

**Questions:**

- **Regarding the performance trade-off:** Have you considered experimenting with parameter-efficient fine-tuning (PEFT) methods, such as LoRA? These techniques modify a smaller subset of parameters and are sometimes known to mitigate catastrophic forgetting of pre-trained abilities. Could such an approach lessen the observed performance drop on answerable questions?

- **Regarding the SFT paradigm:** How do you position your data-centric approach relative to emerging non-invasive methods that detect unanswerability by probing a model's internal states (e.g., using a linear classifier on hidden activations)? Could FactGuard-Bench serve as a high-quality, challenging dataset to train or validate these lightweight classifiers, potentially offering a more efficient solution?

- **Regarding synthetic data artifacts:** The entire benchmark was generated using a single model (Qwen2.5-72B-Instruct). Are you concerned that models fine-tuned on this data might overfit to stylistic quirks or biases of the generator model, thus limiting generalization to human-written questions or questions from other models? Have you considered using a mixture of generator models to enhance the diversity and robustness of the benchmark?

---

> ### Author Response · Authors · 2025-11-21
>
> We appreciate your insightful comments and constructive suggestions, which are indispensable to improving the quality of our work. Below we provide a comprehensive response to your comments, including new experimental evidence, and we hope that they alleviate your previous concerns.
>
> # Responses to Weaknesses and Questions
>
> ## **W1&Q1: Significant Performance Trade-off**
> We provide new evidence to demonstrate that this trade-off can be mitigated.
>
> ### **1.1. In-Domain Performance Stability**
> The fine-tuning does **not** cause "overly cautious" behavior on our **in-domain** test set. As shown in our original Table 4, the accuracy for answerable questions in most experiments **remains at the same high level** or is even slightly improved.
>
> ### **1.2. Mitigation through $\text{LoRA}$ (New Evidence)**
> To rigorously address the concern of catastrophic forgetting, we conducted **Parameter-Efficient Fine-Tuning using LoRA** experiments. The results in the new Table 8 and 9 confirm the effectiveness of LoRA.
>
> To emphasize, the prediction accuracy on the SQuAD 2.0 dev set is as follows:
> | Model | Overall Acc. | Answerable Acc. | Unanswerable Acc. |
> | :--- | :---: | :---: | :---: |
> | Qwen2.5-32B-Instruct | 78.66 | 94.43 | 62.93 |
> | Qwen2.5-32B-Instruct-sft | 79.47 | 90.55 $\downarrow$ | 68.41 $\uparrow$ |
> | **Qwen2.5-32B-Instruct-lora** | **82.73 $\uparrow$** | **95.95 $\uparrow$** | **68.98** $\uparrow$ |
>
> The LoRA results for Qwen2.5-32B demonstrate that PEFT successfully enhances the model's ability to handle unanswerable questions while **simultaneously improving performance on answerable questions** in cross-domain scenarios (94.43% $\to$ 95.95%).  This demonstrates the potential of FactGuard-Bench as **high-quality data** that can be efficiently used to train reliable models.
>
> ### **1.3. Rejection Tendency as Learned Caution**
> The results from our training runs using different ratios of answerable to unanswerable samples provide clear confirmation of this trend. The results can be found in Figure 7.
>
> ### **1.4. Domain Shift**
> The FactGuard-Bench dataset was constructed solely from the legal and book domains due to copyright restrictions.
>
> **Further details on the performance degradation for cross-domain answerable questions can be found in Appendix D.**
>
> ## **W2: Narrow Definition of Unanswerability**
>
> FactGuard-Bench is specifically designed to address the unique challenges of LLMs in long-context reading comprehension and to target hallucination mitigation. While AbstentionBench evaluates LLMs as **"Knowledgeable and Cautious Agents"**, FactGuard-Bench serves as an industry-grade benchmark that assesses them as **"Careful Readers"**. The two are complementary in evaluating overall LLM reliability, with FactGuard-Bench specifically filling the critical gap for large-scale, long-context, and context-faithful evaluation.
>
> ## **W3&Q2: Regarding the SFT Paradigm**
>
> Our data-centric approach presents a substantially greater challenge for the LLM, requiring it not only to recognize answerability but also to produce a well-reasoned explanation.
>
> FactGuard-Bench can serve as a high-quality and challenging dataset for training or validating lightweight classifiers. Because the unanswerable responses in the benchmark contain explicit refusal phrases—such as "The article does not mention...", "This document is not provided...", or "The question cannot be answered..."—which serve as clear signals for detecting whether a model considers a question answerable.
>
>
> ## **Q3: Synthetic Data Artifacts (Single Generator Model)**
> We mitigated this risk through a multi-stage workflow:
>
>  **Evidence-Driven Design:** Our answerable questions are **strictly based on explicit evidence** extracted from the original texts, constraining the generated content to be contextually faithful rather than stylistically dependent on the generator.
>
>  **External Constraints:** The use of RAG to retrieve relevant passages and the World Wide Web to filter out common knowledge conflicts introduced **external, non-Qwen stylistic constraints**, further ensuring the integrity of the questions.
>
> **Diversity Prompts:** We deliberately used diverse prompts to maximize the variety of roles, question styles, and logical patterns for unanswerable responses.
>
> The primary goal of FactGuard-Bench is to test **contextual fidelity** and **long-range reasoning**, not to mimic human question or answer style. Therefore, both single-generative models and multiple-generative models are acceptable.

---

### Author Response · Authors · 2025-12-01
**Summary of Rebuttal & Key Revisions: Mitigation of Trade-offs, Enhanced Rigor, and Benchmark Uniqueness**

Dear Area Chair and Reviewers,

We sincerely thank the reviewers for their valuable feedback. We are encouraged by the positive recognition of our work's **novelty** and **timeliness** in addressing the critical challenge of LLM hallucination in **long-context reading comprehension**.

Our rebuttal directly addresses the core concerns about performance trade-offs, methodological rigor, and the benchmark's scope. We have strengthened our case with **new experiments** and **comprehensive explanations** in **Appendix D**.

---

### **1. Performance Trade-off Mitigation via PEFT (Addressing HKQn W1/Q1, YG22 W3, 3jaj Q3)**

Reviewers identified a critical performance trade-off: SFT improved refusal capabilities but sometimes reduced accuracy on answerable questions.

* **New Evidence:** To mitigate this, we conducted **PEFT using LoRA**.
* **Result (Qwen2.5-32B on cross-benchmark SQuAD 2.0):**

    * **Baseline SFT:** Answerable Acc. decreased from **94.43%** to $\text{90.55\%}$ ($\downarrow$).

    * **LoRA PEFT:** Answerable Acc. **improved** to **95.95%** ($\uparrow$), while unanswerable performance was also significantly enhanced.
* **Conclusion:** This decisively demonstrates that the performance drop is a **manageable capacity trade-off**, not an inherent data flaw. The strategic use of LoRA with larger models provides a viable mitigation strategy. Crucially, this confirms **FactGuard-Bench's potential as a high-quality dataset** that can be efficiently used to train reliable models without compromising performance on answerable queries.

### **2. Enhanced Generalizability and Benchmark Uniqueness (Addressing YG22 W2/Q1, HKQn W2)**

Concerns were raised about generalization and the scope of unanswerability (contextual fidelity).

* **New Evidence:** We added an evaluation on the **KUQA dataset**, serving as a further cross-domain test.
* **Result (Qwen2.5-7B on cross-benchmark KUQA):** Performance on unanswerable questions improved from $\text{60\%}$ (Baseline) to $\mathbf{95\%}$ (SFT). While the KUQA dataset is small (320 samples), this result is consistent with findings on SQuAD 2.0.
* **Benchmark Uniqueness:** As summarized in the new **Table 10 (Appendix)**, FactGuard-Bench uniquely addresses the intersection of **Long-context** and providing a **Ground Truth with Reasoned Refusal** at **Large-scale** with **Low Human Cost**. FactGuard-Bench is an indispensable, **industry-grade benchmark** for evaluating LLMs as **"Careful Readers"** in long-context scenarios, filling a critical gap in the community's study of unanswerable questions.

### **3. Methodological Rigor and Transparency (Addressing 3jaj W1/W2, YG22 W1/Q3/Q4/Q5)**

Reviewers requested more detail on the LLM-centric methodology and the human review process.

* **LLMs as Generators/Evaluators:** We clarified that LLM synthesis is essential for achieving the required **scale and complexity** (25K examples up to 128K context), which manual annotation cannot afford.
* **Human Review & Reliability:** The resulting **Fleiss' Kappa ($\kappa = 0.64$)** reached substantial agreement, which is competitive with similar state-of-the-art annotation efforts, validating the reliability and quality of our data.
* **Prompting and Details:** We commit to making all **prompts, input/output formats, and data details publicly available** upon release of the codebase to ensure full reproducibility.

---

We believe the comprehensive revisions, particularly the new **LoRA experiments**, and the detailed methodological clarifications, have substantially strengthened the paper. FactGuard-Bench presents the first large-scale, long-context benchmark for assessing and improving LLM refusal capabilities.

We hope this summary will quickly assist the Area Chair in grasping the acknowledged value of our work and the concerns we have addressed during the rebuttal.

---

### Meta-Review · Area_Chair_kRSM · 2025-12-18

**Summary:**

This paper proposes a procedure to generate SFT data to train models better at refusing to answer when there's not enough or misleading evidence in the user's provided context. Given seed data, a language model is used to parse documents and generate passages accompanied by answerable and unanswerable questions. Moreover, the unanswerable cases are of different kinds and can be unanswerable due to lack of relevant information in the context or due to misleading contextual information being provided by the user. Authors then proceeds to show that performing post-training in this dataset results in models that more accurately refuse to answer. Reviewers however pointed out that comes at a cost in accuracy in answerable questions, which may be countered by careful regularization as results posted by authors during the discussion phase suggest.

**Reviewer Concerns:**

Reviewers mostly raised concerns regarding the data generation process, as the presentation lacks some details, and quality control measures to ensure the data is free of issues are not super clear. Besides that, reviewers consistently raised questions regarding the effect of post-training on this kind of data in the performance on answerable questions. Authors did respond with extra results due to fine tuning under a PEFT approach, which did alleviate the issue. I tend to side with the more negative leaning reviewers (2/3) and would argue that the presentation lacks detail, but also the evaluation is a bit constrained. A single model class (Qwen2.5 series) is fine-tuned, and it's a bit unclear to what content results generalize. Reviewers also brought up notes on the fact the ability of SFT'ed models to generalise to other domains (i.e., out of the scope of synthetic data generated by a specific language model), and I would add that other details could be clarified, such as the fact that accuracy is confounded by the ability of models to answer independent of context. The benchmark is generated seeded on general knowledge, so that many questions can be answered without any context at all. That should be controlled for by measuring a baseline accuracy without contexts, before and after post-training.

**Reviewer Scores:**

I don't think reviewers would have modified their scores given the replies. The positive leaning reviewer had their concerns partially addressed, but I wouldn't expect a score raise even then, and the more negative leaning reviewers had their concerns only mildly addressed, as issues such as the scope of the evaluation would require major changes on the manuscript, and so would describing the generation process in detail.

I would say this is a generally interesting direction, but the results presentation seem preliminary, and a deeper and broader analysis is needed before acceptance.

---

### Decision · Program_Chairs · 2026-01-26

Reject